# MocapMe: DeepLabCut-Enhanced Neural Network for Enhanced Markerless Stability in Sit-to-Stand Motion Capture

**DOI:** 10.3390/s24103022

**Published:** 2024-05-10

**Authors:** Dario Milone, Francesco Longo, Giovanni Merlino, Cristiano De Marchis, Giacomo Risitano, Luca D’Agati

**Affiliations:** Department of Engineering (DI), University of Messina, Contrada di Dio, 98166 Messina, Italy; francesco.longo@unime.it (F.L.); gmerlino@unime.it (G.M.); cridemarchis@unime.it (C.D.M.); grisitano@unime.it (G.R.); luca.dagati@studenti.unime.it (L.D.)

**Keywords:** human movement analysis, motion tracking, neural network, markerless pose estimation, sit-to-stand analysis

## Abstract

This study examined the efficacy of an optimized DeepLabCut (DLC) model in motion capture, with a particular focus on the sit-to-stand (STS) movement, which is crucial for assessing the functional capacity in elderly and postoperative patients. This research uniquely compared the performance of this optimized DLC model, which was trained using ’filtered’ estimates from the widely used OpenPose (OP) model, thereby emphasizing computational effectiveness, motion-tracking precision, and enhanced stability in data capture. Utilizing a combination of smartphone-captured videos and specifically curated datasets, our methodological approach included data preparation, keypoint annotation, and extensive model training, with an emphasis on the flow of the optimized model. The findings demonstrate the superiority of the optimized DLC model in various aspects. It exhibited not only higher computational efficiency, with reduced processing times, but also greater precision and consistency in motion tracking thanks to the stability brought about by the meticulous selection of the OP data. This precision is vital for developing accurate biomechanical models for clinical interventions. Moreover, this study revealed that the optimized DLC maintained higher average confidence levels across datasets, indicating more reliable and accurate detection capabilities compared with standalone OP. The clinical relevance of these findings is profound. The optimized DLC model’s efficiency and enhanced point estimation stability make it an invaluable tool in rehabilitation monitoring and patient assessments, potentially streamlining clinical workflows. This study suggests future research directions, including integrating the optimized DLC model with virtual reality environments for enhanced patient engagement and leveraging its improved data quality for predictive analytics in healthcare. Overall, the optimized DLC model emerged as a transformative tool for biomechanical analysis and physical rehabilitation, promising to enhance the quality of patient care and healthcare delivery efficiency.

## 1. Introduction

The sit-to-stand (STS) movement is a simple everyday action and a complex biomechanical process that reveals crucial information about an individual’s functional capacity, strength, and balance [1]. It serves as a cornerstone for evaluating the physical abilities and recovery progress across diverse demographic categories, including the elderly and postoperative patients, such as those recovering from surgeries, like total knee arthroplasty [2]. The growing recognition of its diagnostic value has led to a surge of research focus on precise ways to measure and interpret STS mechanics. The methodologies are particularly geared toward its application in clinical environments, where clinicians and healthcare providers aim for precise, actionable insights to inform care decisions. Various sensing technologies and biomechanical strategies have been brought to the forefront to quantify this seemingly simple yet biomechanically complex action. These technologies now not only include traditional marker-based and markerless motion capture systems but have expanded to embrace a broader spectrum, encompassing wearable sensors, inertial measurement units (IMUs), pressure mats, and force platforms [3]. The research landscape in this area is rich and varied, extending from the foundational understanding of effective STS strategies to the application of advanced sensing technologies and further into data interpretation methods that could refine clinical decision-making protocols. In this section, the authors explore some of the seminal works in these areas, highlighting how each contributes to enhancing the accuracy, efficiency, and utility of STS assessments in clinical settings and everyday monitoring.

Despite substantial advancements in markerless motion capture technology, tools such as OpenPose (a cutting-edge software framework designed for real-time multi-person 2D pose estimation using deep learning techniques [4,5,6,7]) exhibit limitations in accurately estimating complex joints, notably the ankle. This is particularly crucial in detailed and dynamic movements, like the sit-to-stand (STS) motion, where capturing precise movement phases is essential for a thorough biomechanical analysis. Recent studies, including the evaluation by Nakano et al. [8], highlight how OpenPose’s accuracy can be compromised by various factors, including the image resolution and joint occlusion. These limitations underscore the imperative to further explore and refine motion capture techniques to ensure precise and reliable measurements, especially in clinical contexts, where such data are critical for functional evaluation and patient rehabilitation.

Starting with the foundational strategies behind STS measurement, the work by Tanaka et al. [9] significantly deepened our understanding by emphasizing the role of momentum. They introduced a markerless motion capture system (MLS) to quantitatively assess the center of gravity (COG) during the STS movement. Their work opened new avenues by offering a cost-effective alternative to traditional marker-based systems (MBSs) and making the technology more accessible for clinical settings. In a parallel advancement, Thomas et al. [10] employed Microsoft’s Azure Kinect as another markerless system to capture cinematic and spatio-temporal variables. Their findings expanded the clinical applicability of STS assessment, even suggesting possibilities for at-home patient evaluation, thereby providing a more comprehensive data set for clinician’s decision-making. While capturing diverse data types is essential, the crux lies in its interpretation to guide clinical decisions. Onitsuka et al. [11] directly addressed this by applying STS mechanics to evaluate recovery patterns in patients post-total knee arthroplasty. Their work correlated certain kinematic strategies with patient-reported outcome measures (PROMs), filling a gap in the scientific framework underlying clinical evaluations. Their methodology could serve as a new indicator for assessing functional recovery post-surgery.

As the focus shifts toward making these assessments part of routine healthcare, especially for vulnerable populations like the elderly, the need for efficient, accessible, and user-friendly technologies becomes imperative. Frailty in older adults is defined as “a clinical syndrome characterized by significant vulnerability resulting from diminished strength, endurance, and physiological function, increasing the risk of adverse health outcomes.” This highlights the importance of early detection and intervention in managing age-related physical vulnerabilities [12]. In this context, Bochicchio et al. [13] presented a novel approach that employs a 3D wearable inertial sensor for estimating muscle power in the elderly during an STS test. Their work is an efficient and practical alternative to the traditional, more cumbersome laboratory-based assessments. Along similar lines, Cobo et al. [14] developed an automated system tailored to the 30 s chair stand test (CST), employing low-cost sensors and a user-friendly Android app to facilitate unsupervised home use. Such advancements are crucial for detecting frailty and other age-related physical vulnerabilities early. Further emphasizing STS importance, van et al. [15] compared instrumented sit-to-stand (iSTS) tests with traditional manual methods, highlighting the superior correlation of iSTS tests with the health status and functional activities in elderly populations. Their research accentuated the significance of evaluating the dynamic phases of the STS tests, which could have broad implications for fall prevention programs. In the quest for more granular biomechanical insights, Howarth et al. [16] provided a deep dive into the kinematic aspects of repeated STS cycles. Their study revealed that the joint angles in the sagittal plane during initial cycles of a 5 × STS test could represent those in isolated STS movements, thus providing a deeper understanding of physical functionality.

Given these challenges, our study aimed to develop an innovative application that leverages camera-based motion capture technology to predict clinically significant movements, such as STS, with enhanced accuracy. By transcending the limitations of existing markerless motion capture systems, including OpenPose, we endeavored to furnish a practical and intuitive tool that enables more timely and informed clinical decisions.

This work aimed to develop an innovative application that utilizes camera-based motion capture technology for accurately predicting clinically inspectable movements, such as STS. The authors intended to enhance the assessment of functional capacity, strength, and balance, particularly focusing on the elderly and postoperative patients. The authors aimed to refine the assessment of functional capacity, strength, and balance, explicitly including elderly individuals at risk of dementia-related issues and falls, as well as patients recovering from a range of surgical procedures, with a specific emphasis on orthopedic surgeries, such as hip and knee replacements.

By employing markerless approaches and integrating cutting-edge technology, the goal was to create an efficient tool capable of swiftly analyzing movements. This tool was designed to overcome the limitations of other markerless motion capture systems, such as OpenPose. The remainder of this paper is organized as follows: Section 2 provides a comprehensive review of the existing literature, focusing on the intersection of bioengineering, artificial intelligence, and their applications in motion analysis. Section 3 introduces the foundational concepts and methodologies, detailing the data sources, the structure of the datasets used, and an overview of the ResNet architecture employed in this study. Section 3.3 offers a high-level overview of the innovative MocapMe system, elucidating its key principles and functionalities, delving deeper into the implementation strategy, and highlighting the integration and utilization of OpenPose and DeepLabCut [17,18] technologies within our framework. Section 4 discusses the experimental results, providing a detailed analysis and evaluation. Finally, Section 5 and Section 6 conclude this paper by summarizing the main outcomes and contributions. This section also outlines future research directions, with the aim to further the scope and impact of this study in motion analysis and clinical applications.

## 2. Related Works

The intersecting domains of bioengineering and artificial intelligence have forged new avenues in various applications, most notably in biomechanical analyses and medical applications, especially for what concerns orthopaedic diseases and neurological conditions [19,20,21,22,23]. This burgeoning field focuses on advanced motion-tracking technologies that leverage neural networks, video analysis, and other computational approaches, including sensor technologies. The expansion of computational techniques has been especially notable in human motion analysis, where markerless motion capture (MMC) technologies have gained prominence. MMC technologies aim to make biomechanical analyses more accessible, adaptable, and cost-effective by utilizing deep learning algorithms and video-based techniques. Researchers have leveraged various methodologies and technologies, including DeepLabCut, OpenPose, and AlphaPose.

Focusing on motor control pathologies primarily associated with muscle coordination challenges during movement, this study established an optimal method for synergy extraction in clinical settings. It highlighted the potential for brief sit-to-stand tests to reliably identify muscle synergies, thus facilitating their use in clinical practice for diagnosing and assessing the rehabilitation progress of individuals with motor impairments.

A significant advancement in applying deep learning to bioengineering is the work of Neil Cronin et al. [24]. Their research employed DeepLabCut and GoPro cameras (manufactured by GoPro Inc. in United States) to assess deep water running (DWR) kinematics in hydrotherapy settings. This methodology, especially when locating body landmarks even when challenged by light variations and motion blur, shares a methodological kinship with the work of Boswell et al. [25]. The latter work focused on predicting the knee adduction moment (KAM) in osteoarthritic patients through 2D video analysis, thus showing how deep learning can accurately capture body dynamics.

Regarding accessibility and affordability, Coias et al. [26] proposed a low-cost virtual trainer system to facilitate home rehabilitation for stroke patients. Similarly, Castelli et al. [27] contributed to this space by eliminating the need for physical markers and using single video cameras for 2D gait analysis. Both works aimed to reduce the cost of biomedical research and healthcare provision, although they often need more specifics about their experimental setups and comparisons between used technologies.

Extending the scope beyond clinical settings, Potempski et al. [28] applied biomechanical analysis to artistic domains, like salsa dancing. They employed OpenPose for pose estimation but notably shifted their focus toward rhythm and movement synchronization. Likewise, OpenPose found utility in Trettenbrein’s research [29] in linguistics and gesture studies. These works showcased the adaptability and broader applicability of MMC technologies and deep learning methodologies.

Parkinson’s disease (PD) serves as another critical field of research. Shin et al. [30] and Sato et al. [31] employed video-based analyses to evaluate symptoms like bradykinesia and walking periodicity in PD patients. Shin’s study relied on MobileNetv2-1.0 algorithms for limb tracking, while Sato’s study emphasized stride cadence through OpenPose. These contributed to more quantitative and objective evaluations of PD symptoms. Further, Haberfehlner et al. [32] utilized a random forest model trained on DeepLabCut-extracted coordinates to propose an automated evaluation system for dystonia in cerebral palsy patients. This paves the way for more automated diagnostic systems in neurobiological disorders. The assessment of bradykinesia, particularly in Parkinson’s disease, highlights the complexity of diagnosing and differentiating this condition. Despite advances in technology and methodology, 3D motion capture systems remain the gold standard for objective measurement. This underscores the importance of precise, detailed motion analysis in understanding and managing Parkinsonian syndromes [33].

Within the realm of sports biomechanics, notable contributions were made by Giulietti et al. with their SwimmerNET [34], which aims to estimate a swimmer’s pose underwater. This work finds a complement in the work of Mundt et al. [35], who aimed to generate synthetic 2D videos from 3D motion capture data to overcome data limitations. These works addressed the complexities of athletic performance, showing significant strides in sports science. Along similar lines, Yamamoto et al. [36] and Nakano et al. [8] evaluated OpenPose in different athletic scenarios to underline both the utility and the limitations, particularly in tracking accuracy.

In a broader healthcare setting, Lonini et al. [37] demonstrated the feasibility of DeepLabCut for gait analysis in post-stroke patients, thus reducing the need for specialized equipment. This work dovetailed with that of Ota et al. [38], who contrasted OpenPose with traditional systems like VICON for walking and running analyses on treadmills. In a similar vein, Drazan et al. [39] applied DeepLabCut to study vertical jumping and emphasized the robustness of MMC in biomechanical data capture. The study by Needham et al. [40] explored the limitations and capabilities of OpenPose in tracking an athlete’s center of mass during linear running activities.

Concerning practical applications and accessibility, Aderinola et al. [41] showed the viability of using a single smartphone camera for MMC in measuring jump height. This is in line with Washabaugh et al.’s study [42], which provided a comparative analysis of different pose estimation methods, like OpenPose, Tensorflow MoveNet, and DeepLabCut, in walking kinematics. However, another avenue of application was shown by Kwon et al. [43], who proposed a machine learning framework using 3D pose estimation and walking pattern classifiers, demonstrating its potential in areas like rehabilitation and the early detection of neurological disorders. On a different note, Moro et al. [44] advocated for a transition to markerless systems, which achieved results comparable to traditional marker-based systems while avoiding their limitations, like high costs and unnatural movement constraints.

While these works signify the transformative potential of MMC technologies, they also highlight some limitations, such as the need for more methodological details concerning camera placement and real-time performance. Nonetheless, the collective implications of these studies herald a future where biomechanical analyses and healthcare technologies are more accessible, adaptable, and cost-effective, albeit with room for further exploration and improvement.

## 3. Material and Methods

### 3.1. Data Sources and Structure

The model under investigation was calibrated based on the analysis of videos portraying the sit-to-stand movement, all of which were recorded with the subject’s left side as the reference point. To achieve this analysis, the authors employed data from two distinct collections:**Primary dataset:** This dataset was sourced from an online repository, as presented in a study by Boswell et al. [45]. The dataset consists of 493 videos, originally captured from various perspectives. These videos were subsequently processed to ensure a consistent view of the movement from the subject’s left side.**Supplementary dataset:** Additionally, a second dataset was specifically curated for this research, comprising 48 videos. These videos were evenly distributed between three subjects, all of Italian nationality, aged between 28 and 37 years, with an average age of 33 years.

The primary dataset was gathered from a diverse group of 493 participants who resided across a majority of the United States (U.S.), specifically in 35 distinct states. These individuals had an average age of 37.5 years, with a broad age range from 18 to 96 years. About 54% of the participants were female. Eligibility for the study required participants to meet several criteria: they had to reside in the U.S., be at least 18 years of age, feel confident in their ability to stand up from a chair without using their arms, and ensure another individual was present during the test for safety.

Following comprehensive data cleansing necessitated by the presence of unfocused frames within the video footage, the cohort finalized for analysis comprised 405 individuals. Within this meticulously refined sample, females represented 53%.

#### 3.1.1. Test Characteristics and Participant Details

The sit-to-stand (STS) test was chosen due to its established clinical relevance in analyzing physical function. It is a test deeply associated with the strength and power of the lower limbs and is frequently utilized by clinicians and researchers for assessing physical function.

The participant characteristics encompassed not only demographic data, such as age, gender, height, weight, ethnicity, education, employment, income, marital status, and state of residence, but also encompassed insights into their physical and mental health, as assessed by the PROMIS v.1.2 Global Health Short Form.

#### 3.1.2. Supplementary Dataset Acquisition

To bolster the validity of our motion capture model, the authors augmented our data with an additional 48 videos. Video acquisition was conducted in a setting specifically prepared to capture precise metrics and angles that could serve as a benchmark for the machine learning model.

As shown in Figure 1, the acquisition environment’s setup was designed to ensure a variety of distances between the subject and the video camera (smartphone). Specifically, videos of the subjects were recorded at distances of 2 m, 3 m, 4 m, and 5 m. Furthermore, each subject was instructed to repeat the sit-to-stand movement at different angles relative to the video camera: 0 degrees, 15 degrees, 30 degrees, and 45 degrees.

### 3.2. Detailed Overview of ResNet Architecture

The ResNet (residual network) architecture, which is an innovation in deep learning, provides a profound understanding of convolutional neural networks’ capabilities. ResNet adeptly addresses the vanishing gradient problem, facilitating the training of exceptionally deep networks. This discourse meticulously dissects ResNet’s design, casting light on its foundational principles and the intricate technicalities interwoven within.

ResNet mandates that images be dimensionally divisible by 32 in terms of both height and width. They should also manifest a channel width of 3, signifying the standard RGB (red, green, blue) channels. The adopted input image shape for our analysis was 224×224×3. This discourse begins with an exploration of the initial layers, which incorporated a ‘conv2d’ convolution layer with a 7×7 kernel and stride of 2, followed by a max-pooling mechanism that applied a 3×3 kernel with a stride of 2. Padding, which was introduced at this juncture, ensured the maintenance of the image shape, modifying it to 230×230×3. This nuance led to an output shape of 112×112×64 after the introductory convolutional layer, culminating in a 55×55×3 output after the max-pooling, setting the stage for subsequent operations.

ResNet34, which is a specific ResNet variant, hinges on two pivotal building blocks: the convolutional block and the identity block. The convolutional block necessitates an alteration in the input image’s dimensions. Intricately designed, each block within ResNet34 amalgamates two layers—Conv2d and BatchNormalization—and an activation function. It is imperative to underscore that the original input image is incorporated into the block’s output upon these operations’ completion.

Conversely, ResNet50 integrates both convolutional and identity blocks. Each block is designed with three layers. The identity blocks consistently maintain a stride of one for each step. In contrast, the convolutional blocks have a distinct configuration involving short concatenations after the third layer, thus integrating the original image into the block’s output.

There is a noticeable doubling in channel width, while the input dimensions undergo halving. This adaptation mechanism underpins consistent performance across varying layers. A comprehensive breakdown of this concept is encapsulated in Table 1, detailing the output dimensions at each transitionary stage.

ResNet34, in its design, superimposes layers for each residual function *F* with 3×3 convolutions. Enhancing this structure is an intermediate pooling layer, culminating in a dense layer embedding 1000 neurons, which is a representation of ImageNet classes. In juxtaposition, the deeper ResNet variants, such as ResNet50, ResNet101, and ResNet152, employ a bottleneck strategy. Here, every residual function *F* integrates three consecutively structured layers, constituting 1×1, 3×3, and 1×1 convolutions. The pivotal role of the 1×1 convolutional layer in dimension reduction and restoration warrants special mention.

### 3.3. MocapMe

This methodology section presents the development of MocapMe (see Figure 2), which is a system designed to advance motion analysis by leveraging OpenPose (OP) and DeepLabCut (DLC). Aimed at surpassing traditional motion capture limits, MocapMe integrates these technologies to enhance analysis precision and accessibility.

The system’s methodology is centered around an optimized DLC-based model, beginning with anatomical keypoint detection via OP and subsequent refinement through DLC training. This approach improves keypoint detection’s stability and accuracy, addressing previous systems’ challenges.

MocapMe is distinguished by its adaptability and ease of use, making it suitable for various applications from sports performance to clinical rehabilitation. Training the DLC model with OP data allows MocapMe to offer a refined motion analysis approach.

The development was driven by merging advanced technical features with practicality, emphasizing adaptability, precision, and simplicity. This section will also cover the choice of Python for its programming foundation due to its integration with advanced motion capture technologies, ensuring MocapMe’s contribution to motion analysis innovation.

### 3.4. Implementation Objectives

The core intent underpinning the presented implementation revolves around effectively leveraging the robust capabilities of both OpenPose and DeepLabCut. The vision was to capitalize on the real-time human pose detection offered by OpenPose and integrate it seamlessly with the detailed and precise keypoint labeling facilitated by DeepLabCut. This confluence not only augmented the analytical precision but also offered a streamlined and largely automated workflow, democratizing the intricacies of motion analysis for a broader audience, irrespective of their depth of expertise in the domain.

Another cardinal objective was the emphasis on system adaptability. It was imperative to engineer a system imbued with a high degree of configurability, ensuring it could be seamlessly tailored to a broad spectrum of applications and use-case scenarios. Integral to this was the system’s ability to accommodate videos with diverse lengths and resolutions, function effectively across a range of camera configurations, and ensure smooth integration compatibility with an array of external libraries and tools.

### 3.5. Implementation Strategy

This approach utilized two pivotal technologies: OpenPose and DeepLabCut. OpenPose, which is a groundbreaking development from Carnegie Mellon University’s Perceptual Computing Lab, is acclaimed for its human pose detection capabilities. By employing deep neural networks, OpenPose accurately identifies human poses in static images and dynamic video sequences. Its proficiency in real-time analysis and multi-subject processing was crucial for capturing extensive motion data.

DeepLabCut, meanwhile, is a specialized framework for motion analysis, leveraging a transfer learning paradigm. This involves a neural network pre-trained on a large dataset, and is later fine-tuned for specific motion analysis tasks. This strategy ensures high precision in keypoint annotations while minimizing training demands, making it highly suitable for detailed motion analysis [18].

Integrating OpenPose and DeepLabCut, we developed a system capable of intricately detecting and analyzing motion within videos. Initially, OpenPose detects human poses in the video, and the extracted data is then refined and formatted for DeepLabCut, which performs detailed motion keypoint annotations. This process results in a comprehensive dataset that delineates the video’s motion dynamics in fine detail.

### 3.6. Implementation Methodology

The implementation strategy is materialized within a Python 3.8.19 class named *Model_Calculation*, which forms the backbone of our analytical pipeline. This class harnesses the capabilities of libraries, such as OpenCV, pandas, and deeplabcut, thereby laying a robust foundation for motion analysis.

The *LoadData* method, which is a crucial component of this class, is responsible for deploying OpenPose on the specified video. It navigates to the OpenPose directory, executes the pose detection algorithm, and processes the JSON outputs to extract key pose information. This information includes 2D and 3D body, facial, and hand keypoints, offering an extensive portrayal of the subject’s movements.

Following this, the *EvaluationDataDeepLabCut* method transforms the output from OpenPose into a format suitable for DeepLabCut. This step involves converting data formats and pruning extraneous data, ensuring the retention of only the most relevant information for detailed motion analysis.

The *DropUnderThreshold* function filters the data based on a predefined confidence threshold, which is a critical step in maintaining the reliability of our motion analysis by focusing on the most dependable keypoints.

For the targeted examination of specific video segments, the *ExtractFrames* function is utilized. This function adeptly isolates and extracts frames of interest, allowing for the focused analysis of key moments or movements within the video.

The heart of our analytical process is the *FeatureExtraction* method. This method integrates the functionalities of the preceding steps, processing the pose data extracted by OpenPose and preparing it for the detailed keypoint annotation via DeepLabCut.

Lastly, the *DeepLabCutModule* function manages the initialization, training, and evaluation of the DeepLabCut model. This comprehensive process ensures that the model is precisely tuned and capable of delivering accurate motion analysis results.

The user interface of our system, which is facilitated by the *algorithm_openpose_deeplabcut.py* module, capitalizes on the *Model_Calculation* class. This module streamlines the analysis of videos, orchestrating the entire data processing workflow and generating the final outputs. The sequential flow and integration of these components are elaborated in Algorithm 1, providing a structured overview of our methodology.

**Algorithm 1** Motion analysis using OpenPose and DeepLabCut.
1:**procedure** MotionAnalysis(video, outputPath, openPosePath, deepLabCutPath)2:    poseData← ExtractPoseData(video,openPosePath)3:    deepLabCutModel← InitializeDeepLabCut(deepLabCutPath)4:    **for** frame in ExtractFrames(video,poseData) **do**5:        keypoints← ExtractKeypoints(frame,deepLabCutModel)6:        SaveOutput(keypoints,outputPath)7:    **end for**8:
**end procedure**
9:**function** ExtractPoseData(video,openPosePath)   ▹ Deploy OpenPose and filter output10:    Process video with OpenPose.11:    Parse and filter pose data based on confidence.12:    **return** Filtered pose data.13:
**end function**
14:**function** InitializeDeepLabCut(path)       ▹ Set up and return DeepLabCut model15:    Initialize and configure the DeepLabCut model.16:    **return** Initialized model.17:
**end function**
18:**function** ExtractFrames(video,data)        ▹ Return frames of interest for analysis19:    Extract frames based on pose data.20:    **return** List of frames.21:
**end function**
22:**function** ExtractKeypoints(frame,model)    ▹ Apply DeepLabCut to extract keypoints23:    Annotate frame with keypoints using the model.24:    **return** Annotated keypoints.25:
**end function**
26:**function** SaveOutput(data,path)             ▹ Save data to specified path27:    Save data.28:
**end function**



### 3.7. Training Methodology and Analytical Outcomes

The training phase represented a pivotal phase of this research, critically honing the model’s proficiency in deciphering the complex motion dynamics. Leveraging the DeepLabCut framework and synergizing with the computational prowess of a ResNet152 architecture, the authors meticulously trained the model on a compilation of videos that predominantly originated from smartphone recordings, which constituted the previously delineated foundational dataset [45]. To augment the diversity and richness of the training corpus, the authors infused additional footage meticulously captured by the research team, thereby enhancing the model’s exposure to a wide array of motion patterns.

#### Data Preparation and Refinement

The preliminary stage entailed a thorough processing of the video data through the OpenPose framework to distill pose-related metrics. This quantified information was systematically encoded into CSV files and subsequently transformed into H5 file formats, priming them for the training exigencies of DeepLabCut. A pivotal element of the research preparatory methodology was the strategic curation of keypoints extrapolated from the COCO model. Authors’ focal keypoints—encompassing the foot, ankle, knee, hip, and shoulder—were selected for their critical bearing on motion analytics; these keypoints were extracted with mathematical precision, assigning X- and Y-coordinates, along with confidence indices for each keypoint *i*, as specified in the ensuing enumeration:X-coordinate column: (i+1)×3−3;Y-coordinate column: (i+1)×3−2;Confidence value column: (i+1)×3−1.

A post-filtering phase was conducted by the exclusion of frames that exhibited sub-98% accuracy. To train the model, the dataset was automatically split by DeepLabCut into 80% for the training set and 20% for the validation set. The training process encompassed multiple iterations, as enumerated in Table 2 and characterized by distinct dataset volumes, shuffle metrics, and precision measurements. The tabulated results encapsulate the training and test errors, presented in pixel units, alongside p-cutoff values, underscoring the empirical rigor of our methodology.

Figure 3 shows the training loss trajectory and the learning rate throughout the training cycles. The training loss, depicted in blue, exhibited a steep decline, which was especially noticeable in the early phases—this is a reflection of the model’s rapid adaptation during the initial training phase. This decline became more gradual as the training progressed.

The learning rate, shown in orange, was maintained at a fixed level, indicating that a constant learning rate was employed during the training. The consistent learning rate, along with the downward trend of the training loss, suggests that the model learned effectively without the need for a dynamic adjustment of the learning rate.

The learning curve and constant learning rate together informed the strategy behind the choice of hyperparameters and the overall approach to training. They highlight the importance of the data preparation phase, during which data points were carefully selected to ensure the highest quality for model training.

## 4. Results

The comprehensive analysis conducted in this study highlighted the significant performance differences between OpenPose and the DeepLabCut-based model MocapMe in motion capture, especially in the STS movement. To rigorously evaluate the network, and hence, the results delineated herein, an additional set of 20 videos was recorded, none of which were part of either the training or validation set. These 20 videos encompassed 10 different angles and distances, capturing the movements of two distinct human subjects. This is substantiated by Figure 4, Figure 5 and Figure 6, which collectively provide a comprehensive understanding of the model’s capabilities.

### 4.1. Computational Efficiency

The computational demands of motion capture technologies are a pivotal consideration in clinical settings. Figure 5 reveals a consistent trend of reduced processing times for MocapMe compared with OpenPose. This efficiency stemmed from MocapMe’s streamlined algorithmic structure, which optimizes neural network operations, facilitating a rapid analysis of STS movements without compromising the pose estimation accuracy.

### 4.2. Reliability and Precision of Motion Tracking

A central aspect of markerless motion analysis is the consistency with which keypoints are estimated (i.e., confidence) and their tracking precision.

#### 4.2.1. Reliability in Keypoint Estimation

The bar chart in Figure 4 encapsulates the aggregate confidence as a measure of reliability across all videos for five selected keypoints. It is evident that MocapMe consistently maintained a higher average confidence level than OpenPose, signifying more reliable detection across the datasets. MocapMe demonstrated remarkable stability in the confidence scores, maintaining a tight range with minimal fluctuations. In contrast, OpenPose displayed a broader range of confidence scores, indicating a variance in the detection reliability (Figure 4). Such variability could stem from diverse factors, such as changes in lighting, occlusion, and subject movement complexity. The insights drawn from these analyses validate the robustness of MocapMe in delivering consistent and reliable keypoint detections.

#### 4.2.2. Precision in Keypoint Tracking

Figure 6 illustrates the distribution of distances from the centroid across video frames for the ankle and foot keypoints. MocapMe exhibited a dense clustering of data points, suggesting a higher fidelity in capturing the kinematics of STS movements and more stable keypoint tracking. For each video, the ankle and foot keypoints were treated as stationary points; due to the absence of a gold standard measurement (i.e., marker-based stereophotogrammetric system), this stability parameter was only calculated for these keypoints, which were assumed to be stable during the whole STS motion. These results corroborate the superior reliability and robustness of MocapMe with respect to OpenPose.

## 5. Discussion

This paper presents MocapMe, which is a DeepLabCut-based and OpenPose-informed approach for markerless motion capture. The proposed methodological approach integrates the advantages of the two platforms, showing how a filtered OpenPose-based labeling for DeepLabCut network training yields an improved motion tracking reliability and robustness when compared with standard OpenPose alone. The present study tested the methodological approach on a sit-to-stand scenario, in which the network was trained by using data from an open dataset and was later tested on ad hoc sit-to-stand videos recorded with different camera orientations and distances. The analysis conducted in this study highlighted the performance advantages of the MocapMe trained model (employing DeepLabCut) over OpenPose in markerless motion capture with reference to the context of sit-to-stand movements. The results underscore the enhanced reliability of the MocapMe model in capturing the complex dynamics of STS movements; all the considered keypoints (shoulder, hip, knee, ankle, and foot) were tracked with a significantly higher confidence when compared with OpenPose alone, with confidence levels constantly higher than 0.95. The MocapMe performance was also higher in terms of precision, as calculated from the tracking stability of the stationary ankle and foot keypoints during the STS motion. These joints play a crucial role in the functional assessment of STS movements, and the improved tracking accuracy at the ankle–foot level is particularly significant since small errors in these anatomical landmark localization could affect the whole kinematic chain. The findings from this study suggest the potential use of MocapMe in rehabilitation protocols and patient monitoring systems. Moreover, the consistency and reliability of MocapMe in keypoint detection, as demonstrated in the comparative analysis of confidence levels, reinforce its suitability for future applications to clinical gait analysis and rehabilitation monitoring. Additionally, this study demonstrated a marked improvement in computational efficiency when using MocapMe compared with OpenPose. The reduced processing times, as depicted in Figure 4, are indicative of MocapMe’s streamlined algorithmic structure, which not only accelerated the analysis of STS movements but did so without compromising the accuracy of the pose estimation. This efficiency is of paramount importance in clinical settings, where time efficiency aligns with the fast-paced nature of clinical workflows and the need for rapid, accurate assessments. This study had various limitations since it focused on the tracking of a limited set of keypoints, it only analyzed the performance on 2D images, and it did not compare the results with gold standard measurements coming from marker-based stereophotogrammetric systems. These limitations suggest avenues for future research. As a matter of fact, expanding the scope of analysis to include other joints and movements could provide a more comprehensive understanding of human movement biomechanics. Future developments of MocapMe will allow for the bilateral tracking of an extended set of anatomical landmarks, also including the trunk and the head. Moreover, accurate and reliable tracking from single-camera 2D videos, as emerges from the presented results, paves the way for a markerless 3D motion capture, where the accurate tracking of single keypoints by more than one camera is a fundamental element. Exploring the integration of MocapMe with real-time monitoring systems offers possibilities for advancing dynamic motion analysis in clinical and rehabilitative settings.

## 6. Conclusions

This study highlighted the distinct performance advantages of the (DeepLabCut-based and OpenPose-informed) MocapMe trained model over OpenPose alone in markerless motion capture, especially in the context of sit-to-stand movements. The results notably underscore the enhanced consistency of the MocapMe model in capturing the complex dynamics of STS movements for different anatomical landmarks, with a specific focus on the ankle and foot keypoints. Additionally, this study demonstrated a marked improvement in computational efficiency when using MocapMe compared with OpenPose. The reduced processing times were indicative of MocapMe’s streamlined algorithmic structure, which not only accelerated the analysis of STS movements but did so without compromising the accuracy of the keypoint tracking. The findings from this study advocate for the future integration of MocapMe into rehabilitation protocols and patient-monitoring systems. Its efficiency and accuracy in data capture could support the assessment of patient mobility and balance, which are critical aspects in postoperative recovery and geriatric care. Future research could expand the scope of the analysis to include other joints and movements to provide a more comprehensive understanding of human biomechanics. Additionally, exploring the integration of MocapMe with real-time monitoring systems offers exciting possibilities for advancing dynamic motion analysis in clinical and rehabilitative settings.

## Figures and Tables

**Figure 1 sensors-24-03022-f001:**
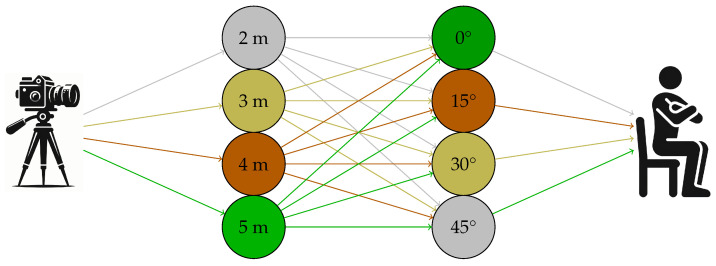
Schematic representation of distances and angles.

**Figure 2 sensors-24-03022-f002:**
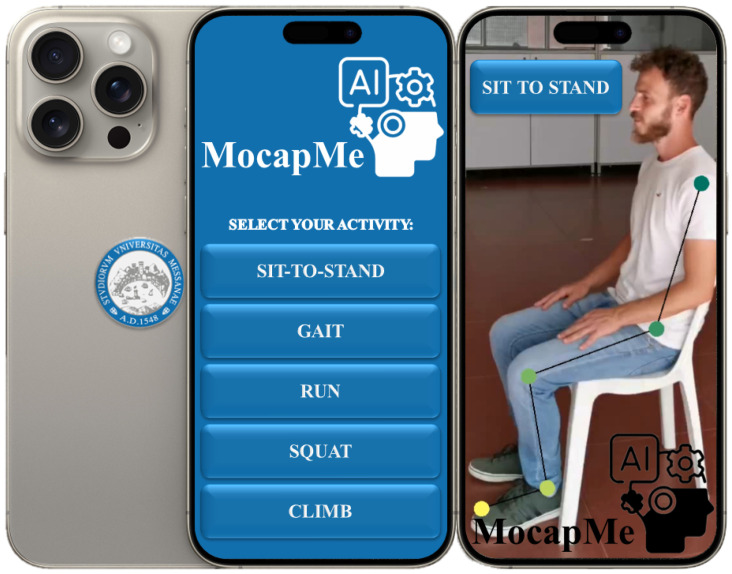
MocapMe mobile application snapshot.

**Figure 3 sensors-24-03022-f003:**
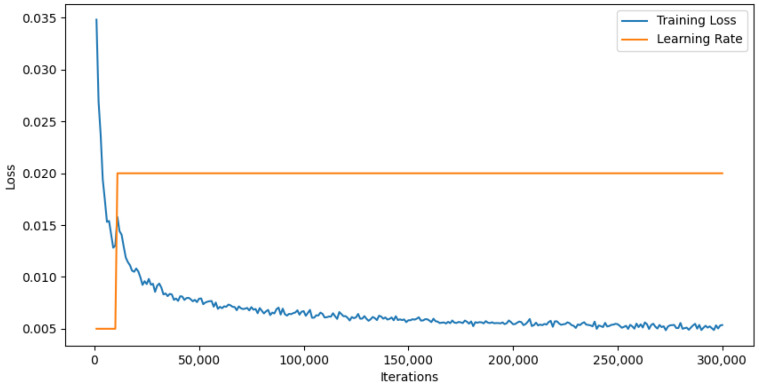
Training loss and learning rate over iterations, illustrating the model’s learning process. The blue trajectory delineates the training loss, indicating a significant decrease as the iterations progressed, which demonstrated the model’s capacity to learn effectively. The orange line represents the learning rate, which remained constant throughout the training process.

**Figure 4 sensors-24-03022-f004:**
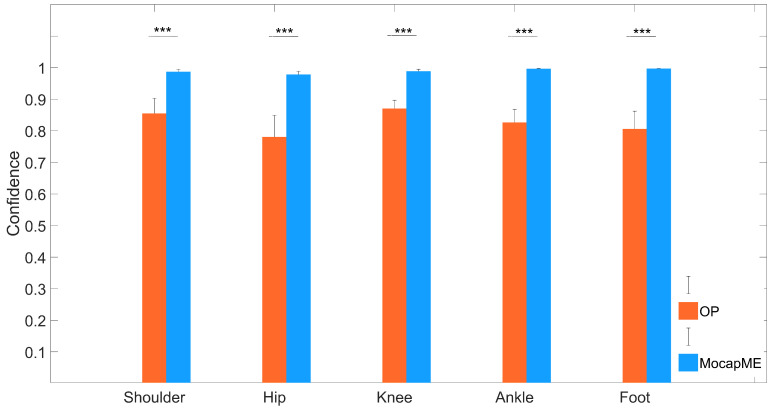
Confidence (mean ± std) of the selected keypoints for OpenPose and MocapMe (DeepLabCut-based). Each bar plot corresponds to one of the five points, showcasing the models’ performance consistencies across the dataset. *** indicates $p<10^{-10}$.

**Figure 5 sensors-24-03022-f005:**
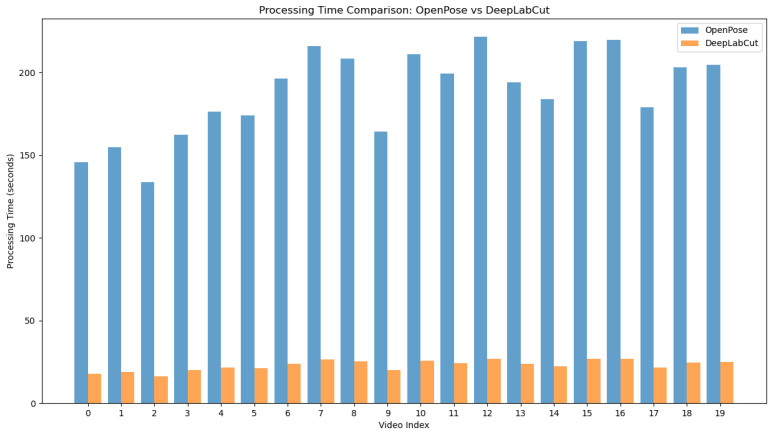
Processing time comparison between OpenPose and DeepLabCut-based MocapMe across various videos, underscoring the enhanced efficiency of MocapMe.

**Figure 6 sensors-24-03022-f006:**
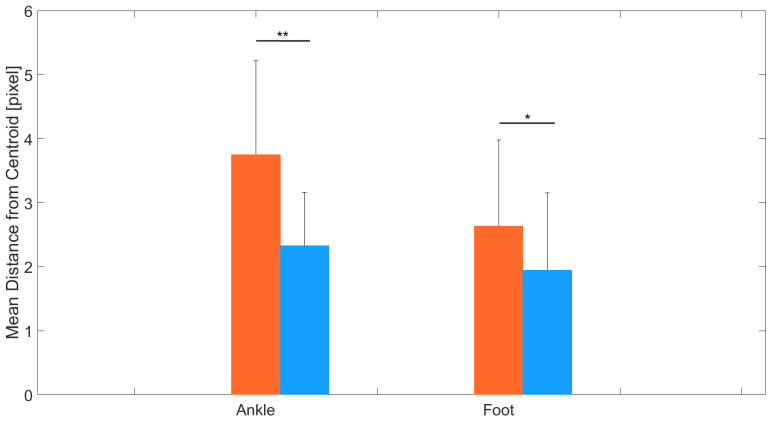
Stability (mean ± std) of the ankle and foot keypoints for OpenPose and MocapMe (DeepLabCut-based). Each bar plot shows the model performance in terms of the mean distance from the centroid. ‘*’ indicates p<0.005, ‘**’ indicates p<0.0005.

**Table 1 sensors-24-03022-t001:** Comparison of different neural network models based on the number of layers.

Layer Name	Output Size	18-Layer	34-Layer	50-Layer	101-Layer	152-Layer
conv1	112 × 112	7 × 7, 64, stride 2				
		3 × 3 max pool, stride 2				
conv2_x	56 × 56	$\left[\begin{array}{c} 3\;\times \;3,64\\ 3\;\times \;3,64 \end{array}\right]$× 2	$\left[\begin{array}{c} 3\;\times \;3,64\\ 3\;\times \;3,64 \end{array}\right]$× 3	$\left[\begin{array}{c} 1\;\times \;1,64\\ 3\;\times \;3,64\\ 1\;\times \;1,256 \end{array}\right]$× 3	$\left[\begin{array}{c} 1\;\times \;1,64\\ 3\;\times \;3,64\\ 1\;\times \;1,256 \end{array}\right]$× 3	$\left[\begin{array}{c} 1\;\times \;1,64\\ 3\;\times \;3,64\\ 1\;\times \;1,256 \end{array}\right]$× 3
conv3_x	28 × 28	$\left[\begin{array}{c} 3\;\times \;3,128\\ 3\;\times \;3,128 \end{array}\right]$× 2	$\left[\begin{array}{c} 3\;\times \;3,128\\ 3\;\times \;3,128 \end{array}\right]$× 4	$\left[\begin{array}{c} 1\;\times \;1,128\\ 3\;\times \;3,128\\ 1\;\times \;1,512 \end{array}\right]$× 4	$\left[\begin{array}{c} 1\;\times \;1,128\\ 3\;\times \;3,128\\ 1\;\times \;1,512 \end{array}\right]$× 4	$\left[\begin{array}{c} 1\;\times \;1,128\\ 3\;\times \;3,128\\ 1\;\times \;1,512 \end{array}\right]$× 8
conv4_x	14 × 14	$\left[\begin{array}{c} 3\;\times \;3,256\\ 3\;\times \;3,256 \end{array}\right]$× 2	$\left[\begin{array}{c} 3\;\times \;3,256\\ 3\;\times \;3,256 \end{array}\right]$× 6	$\left[\begin{array}{c} 1\;\times \;1,256\\ 3\;\times \;3,256\\ 1\;\times \;1,1024 \end{array}\right]$× 6	$\left[\begin{array}{c} 1\;\times \;1,256\\ 3\;\times \;3,256\\ 1\;\times \;1,1024 \end{array}\right]$× 23	$\left[\begin{array}{c} 1\;\times \;1,256\\ 3\;\times \;3,256\\ 1\;\times \;1,1024 \end{array}\right]$× 36
conv5_x	7 × 7	$\left[\begin{array}{c} 3\;\times \;3,512\\ 3\;\times \;3,512 \end{array}\right]$× 2	$\left[\begin{array}{c} 3\;\times \;3,512\\ 3\;\times \;3,512 \end{array}\right]$× 3	$\left[\begin{array}{c} 1\;\times \;1,512\\ 3\;\times \;3,512\\ 1\;\times \;1,2048 \end{array}\right]$× 3	$\left[\begin{array}{c} 1\;\times \;1,512\\ 3\;\times \;3,512\\ 1\;\times \;1,2048 \end{array}\right]$× 3	$\left[\begin{array}{c} 1\;\times \;1,512\\ 3\;\times \;3,512\\ 1\;\times \;1,2048 \end{array}\right]$× 3
	1 × 1	Average pool, 1000-d fc, softmax				
FLOPs		$1.8\times 10^{9}$	$3.6\times 10^{9}$	$3.8\times 10^{9}$	$7.6\times 10^{9}$	$11.3\times 10^{9}$

**Table 2 sensors-24-03022-t002:** Training results of DeepLabCut model.

Iter.	Train Iter.	Dataset (%)	Shuffle	Train Error (px)	Test Error (px)	p-Cut	Train Err (p-Cut)	Test Err (p-Cut)
0	200 k	80	1	10.7	10.6	0.6	6.37	6.47
1	250 k	80	1	11.45	11.41	0.6	7.89	7.68
2	300 k	80	1	10.26	10.25	0.6	6.41	6.47

## Data Availability

The data presented in this study are available on request from the corresponding author.

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
