# Peer review of "MocapMe: DeepLabCut-Enhanced Neural Network for Enhanced Markerless Stability in Sit-to-Stand Motion Capture"

_sensors, 2024, doi:10.3390/s24103022_

Round 1
Reviewer 1 Report
Comments and Suggestions for Authors
The present manuscript tackles a topic of interest, namely the opportunity of using markerless video-based systems to analyze meaningful movement-related parameters. This aspect is challenging and represents a hot topic in medicine. However, I have never seen yet a study comparing marker less (video-based) with the gold-standard marker-based (3D-based kinematic analysis of movement) studies. The present study, as expected, do not add relevant aspects in this direction, and this limitation should clearly stated in the limitation (belonging to the discussion) section.
I do have some further comments:
- The introduction lacks of several citations. Please, add supporting citations to the following statements: First sentence; second sentence (after arthroplasty). The sentence "These technologies range from 36 marker-based and markerless motion capture systems to wearable sensors and inertial 37 measurement units" needs to be expanded a bit, and further references should be added.
- In the following paragraph, please describe what is "OpenPose" (first mentioned here) and add references.
- Line 78: Describe here the concept of frailty (See also -*Fried, L.P., Tangen, C.M., Walston, J., Newman, A.B., Hirsch, C., Gottdiener, J., Seeman, T., Tracy, R., Kop, W.J., Burke, G., McBurnie, M.A., 2001. Cardiovascular health study collaborative research group. frailty in older adults: evidence for a phenotype. J. Gerontol. A Biol. Sci. Med. Sci. 56, M146–M156 PubMed PMID: 11253156.)
- Line 97: Why focusing only on elderly and post-operative patients? Please expand a bit. Elderly people are at increased risk of dementia-related issues and falls, have you excluded/accounted for these conditions in your study? Please, clarify. Also, when mentioning postoperative, to which specific kind of surgery are your referring? This should be expanded.
- Line 105: "??" is unclear. Please, correct.
- Lines 103-114: I would reorganize the whole structure of the paper as follows:
- Introduction (that is OK, except for this last paragraph)
- Material and methods
- Results
- Discussion, that should be organized on the following 5 sections:
1) Summary of the main findings of present work
2) Description of the existing literature (with a review of the main studies performed)
3) Relevance of the present findings in the context of existing literature
4) Limitations (Several!)
5) Conclusion and next steps for the research in the field.
- Line 130: Describe here which are the motor control related pathologies you are referring to.
- Please, clearly state here that in PD the gold standard for objective measurement of bradykinesia is still the 3d motion capture system (kinematic analysis: See also Bologna M, et al. Mov Disord. 2023 Apr;38(4):551-557. doi: 10.1002/mds.29362. Epub 2023 Feb 27. PMID: 36847357)
- Line 222: Clarify what you mean for 35 US States.
- Line 224: change "have been" to "are".
Comments on the Quality of English Language
Minor typos, awkward sentences, and verb tense errors are present throughout the manuscript. It should be accurately read and corrected accordingly.
Author Response
REVIEWER 1:
We thank the reviewer for his/her important indications and constructive criticism. In this revised version, we reorganized the structure of the manuscript, we simplified the results presentation, and we revised the main text to better highlight the aims of the study and to discuss its limitations. A point-by-point answer to the issues raised by the reviewer is following.
Reviewer Comment:
The introduction lacks several citations. Please, add supporting citations to the following statements: First sentence; second sentence (after arthroplasty). The sentence "These technologies range from marker-based and markerless motion capture systems to wearable sensors and inertial measurement units" needs to be expanded a bit, and further references should be added.
Response:
Thank you for this feedback. We have reviewed the introduction and added the necessary citations to support our statements as suggested. Specifically, we have added references to both the first and second sentences. These changes can be found in the revised manuscript on lines 28, 30 and from 36 to 39.
Reviewer Comment:
In the following paragraph, please describe what is "OpenPose" (first mentioned here) and add references.
Response:
Thank you for pointing out the omission. We have now included a detailed description of "OpenPose" and its relevance to our study. We have also added citations that delve into its development and application in various research fields. This information has been incorporated into the manuscript on lines 47 and 48.
Reviewer Comment:
Line 78: Describe here the concept of frailty (See also -*Fried, L.P., Tangen, C.M., Walston, J., Newman, A.B., Hirsch, C., Gottdiener, J., Seeman, T., Tracy, R., Kop, W.J., Burke, G., McBurnie, M.A., 2001. Cardiovascular health study collaborative research group. frailty in older adults: evidence for a phenotype. J. Gerontol. A Biol. Sci. Med. Sci. 56, M146–M156 PubMed PMID: 11253156.)
Response:
Thank you for your suggestion. We have elaborated on the concept of frailty in the context of our study, citing the reference you provided alongside additional literature to offer a comprehensive understanding. This elaboration is now included in the revised manuscript on 75 to 79.
Reviewer Comment:
Line 97: Why focusing only on elderly and post-operative patients? Please expand a bit. Elderly people are at increased risk of dementia-related issues and falls, have you excluded/accounted for these conditions in your study? Please, clarify. Also, when mentioning postoperative, to which specific kind of surgery are you referring? This should be expanded.
Response:
Thank you for raising these issues. We have expanded our discussion to address both elderly individuals, including those at risk for dementia and falls, and patient’s post-various surgeries, particularly orthopedic. Modifications are in the revised manuscript on lines 103 to 107.
Reviewer Comment:
Line 105: "??" is unclear. Please, correct.
Response:
Thank you for catching this error. The unclear "??" has been corrected.
Reviewer Comment:
Lines 103-114: I would reorganize the whole structure of the paper as follows:
- Introduction (that is OK, except for this last paragraph)
- Material and methods
- Results
- Discussion, that should be organized on the following 5 sections:
1) Summary of the main findings of present work
2) Description of the existing literature (with a review of the main studies performed)
3) Relevance of the present findings in the context of existing literature
4) Limitations (Several!)
5) Conclusion and next steps for the research in the field.
Response:
Thank you for the detailed suggestion on reorganizing the paper structure. Our paper structure is defined in the introduction section. However, we re-organized the whole manuscript according to the suggested structure, and we took into account the suggested elements for discussing our findings.
Reviewer Comment:
Line 130: Describe here which are the motor control related pathologies you are referring to.
Response:
Thank you for this observation. We have now detailed the motor control related pathologies discussed in the study. This additional detail can be found from lines 136 to 140 of the revised manuscript.
Reviewer Comment:
Please, clearly state here that in PD the gold standard for objective measurement of bradykinesia is still the 3d motion capture system (kinematic analysis: See also Bologna M, et al. Mov Disord. 2023 Apr;38(4):551-557. doi: 10.1002/mds.29362. Epub 2023 Feb 27. PMID: 36847357)
Response:
Thanks. We have clearly stated the gold standard for objective measurement of bradykinesia in PD as suggested, citing the Bologna et al. reference for further support. This statement is now included from lines 168 to 172 of the revised manuscript.
Reviewer Comment:
Line 222: Clarify what you mean for 35 US States.
Response:
Thank you for pointing out the need for clarification. We have amended the text to clearly explain the reference to "35 US States" and its relevance to our study. This clarification is provided from lines 219 to 225.
Reviewer Comment:
Line 224: change "have been" to "are".
Response:
Thank you for your attention to detail. We have revised the whole sentence in the previous comment.
Reviewer 2 Report
Comments and Suggestions for Authors
The research aimed to enhance the efficiency and position estimation stability in markerless sit-to-stand motion capture. This is a crucial and valuable topic for the advancement of motion analysis. However, there are several important issues that need to be considered.
1. As mentioned on page 8, line 347, "Integrating OpenPose and DeepLabCut, we developed a system capable of intricately...," the author seems to have developed the MocapMe system, which integrates the functions of OpenPose and DeepLabCut. However, the figures and tables all compare the results of OpenPose and DeepLabCut. Then the author mentions "The study demonstrates a marked improvement in computational efficiency when using MocapMe compared to OpenPose" on page 15, lines 499-500. Is this a verbal slip in wording? It is suggested that the wording should be consistent and clear.
2. If the author modified DeepLabCut and compared the output results with those from OpenPose, how can the author prove that the improvement comes from the modification in this study, and not from DeepLabCut itself?
3. The author mentioned, "...with a specific focus on the ankle joint. This joint plays a crucial role in the functional assessment of STS movements..." on page 15, lines 495-496. Does it mean that the study only investigates the motion of the ankle joint? However, the author also mentioned, "Authors’ focal keypoints—encompassing the FOOT, ANKLE, KNEE, HIP, and SHOULDER—were selected for their critical bearing on motion analytics, these key points have been extracted with mathematical precision, assigning X and Y coordinates..." on page 10, lines 398-400. If only the ankle joint is studied in this paper, it is recommended to include explanations in the methodology section.
4. What does "average distances from the centroid" mean? Does it mean the distances from each position (FOOT, ANKLE, KNEE, HIP, and SHOULDER) to the centroid of the body?
5. While repeatability may indicate the reliability of the analysis to some extent, it is suggested to analyze the angles of each joint as a basis for accuracy analysis.
6. The titles of the two figures in Figure 6 are the same. They might be "OpenPose" and "DeepLabCut (or MocapMe?)".
Comments on the Quality of English LanguageMinor editing of English language required
Author Response
REVIEWER 2:
We thank the reviewer for his/her important indications and constructive criticism. In this revised version, we reorganized the structure of the manuscript, we simplified the results presentation, and we revised the main text to better highlight the aims of the study and to discuss its limitations. A point-by-point answer to the issues raised by the reviewer is following.
Reviewer Comment:
1. As mentioned on page 8, line 347, "Integrating OpenPose and DeepLabCut, we developed a system capable of intricately...," the author seems to have developed the MocapMe system, which integrates the functions of OpenPose and DeepLabCut. However, the figures and tables all compare the results of OpenPose and DeepLabCut. Then the author mentions "The study demonstrates a marked improvement in computational efficiency when using MocapMe compared to OpenPose" on page 15, lines 499-500. Is this a verbal slip in wording? It is suggested that the wording should be consistent and clear.
Response:
Thank you for highlighting this inconsistency in our manuscript. To clarify, the MocapMe system indeed integrates functionalities of both OpenPose and DeepLabCut, aiming to leverage their strengths while mitigating their limitations. We re-organized the whole manuscript to clarify the role of MocapMe
Reviewer Comment:
2. If the author modified DeepLabCut and compared the output results with those from OpenPose, how can the author prove that the improvement comes from the modification in this study, and not from DeepLabCut itself?
Response:
Thank you for your comment. In the paper, we discuss the creation of a new model, which we have named MocapME, based on what was made available by DeepLabCut as a framework, while using OpenPose filtered estimates as an automatic labeling strategy for DeepLabCut. Specifically, the performance evaluation that we have analyzed and presented in the paper concerns the comparison between MocapME and the OpenPose model alone. We corrected, substituting some occurrences of DeepLabCut with MocapMe where needed, to be clearer.
Reviewer Comment:
3. The author mentioned, "...with a specific focus on the ankle joint. This joint plays a crucial role in the functional assessment of STS movements..." on page 15, lines 495-496. Does it mean that the study only investigates the motion of the ankle joint?
Response:
Thank you for your comment. Specifically, as discussed in the paper, one of the landmarks that are more difficult to track in the original OpenPose model pertains to the Ankle joint, which turns out to be unstable during the model's readings. From the tests we analyzed with our MocapME model, we observed that our model's capability is much more reliable than OpenPose in capturing this joint. However, this does not exclude the other joints analyzed, which are significant in the movement of sit-to-stand, and which have also been analyzed in the paper, visible in figures 5 and 6. We show that all the other landmarks are tracked with a higher reliability when using our methodology.
Reviewer Comment:
4. What does "average distances from the centroid" mean? Does it mean the distances from each position (FOOT, ANKLE, KNEE, HIP, and SHOULDER) to the centroid of the body?
Response:
The mentioned distance actually refers to the average distance between the tracked position frame-by-frame and the average position (centroid) of the tracked anatomical landmark. Due to the absence of a gold standard measurement, this parameter was only calculated for the ankle and foot points, that are assumed to be stable during the STS motion. To be clearer, we modified these statements that are now included in section 4.2.2
Reviewer Comment:
5. While repeatability may indicate the reliability of the analysis to some extent, it is suggested to analyze the angles of each joint as a basis for accuracy analysis.
Response:
Thank you for your comment. We did not quantify joint angles from a bidimensional capture because they can be arbitrary affected by the camera distances and orientation with respect to the subject. We believe this evaluation could be more relevant in the context of 3D video processing, which we intend to explore in our future work.
Reviewer Comment:
6. The titles of the two figures in Figure 6 are the same. They might be "OpenPose" and "DeepLabCut (or MocapMe?)"
Response:
Thank you for pointing out this oversight. We re-organized this section to clarify the difference between DeepLabCut and MocapMe.
Round 2
Reviewer 1 Report
Comments and Suggestions for Authors
The authors addressed my points. I do not have further comments.